# West Nile Virus Seroprevalence in Wild Birds and Equines in Madrid Province, Spain

**DOI:** 10.3390/vetsci11060259

**Published:** 2024-06-07

**Authors:** Richard A. J. Williams, Hillary A. Criollo Valencia, Irene López Márquez, Fernando González González, Francisco Llorente, Miguel Ángel Jiménez-Clavero, Núria Busquets, Marta Mateo Barrientos, Gustavo Ortiz-Díez, Tania Ayllón Santiago

**Affiliations:** 1Department of Genetics, Physiology and Microbiology, Faculty of Biology, Complutense University of Madrid, José Antonio Nováis, 28040 Madrid, Spain; 2Faculty of Health Sciences, Alfonso X El Sabio University, 28691 Madrid, Spain; hillcriova@gmail.com; 3Group for the Rehabilitation of Native Fauna and their Habitat—GREFA, 28220 Madrid, Spain; irene@grefa.org (I.L.M.); fgonzalez@grefa.org (F.G.G.); 4Department of Pharmacology and Toxicology, Faculty of Veterinary Medicine, Complutense University of Madrid, 28040 Madrid, Spain; 5Animal Health Research Center (CISA-INIA), CSIC, 28130 Valdeolmos, Spain; dgracia@inia.csic.es (F.L.);; 6IRTA, Animal Health Program, Animal Health Research Center (CReSA), Campus of the Autonomous University of Barcelona (UAB), 08193 Cerdanyola del Vallès, Spain; 7Mixed Research Unit IRTA-UAB in Animal Health, Animal Health Research Center (CReSA), Campus of the Autonomous University of Barcelona (UAB), 08193 Cerdanyola del Vallès, Spain; 8Department of Microbiology and Parasitology, Faculty of Pharmacy, Complutense University of Madrid, 28040 Madrid, Spain; mmateo14@ucm.es; 9Department of Animal Medicine and Surgery, Faculty of Veterinary Medicine, Complutense University of Madrid, 28040 Madrid, Spain; gusortiz@ucm.es

**Keywords:** flavivirus, West Nile virus, birds, equines, seroprevalence, central Spain

## Abstract

**Simple Summary:**

West Nile virus (WNV) is a flavivirus that circulates among birds and mosquitoes and can cause outbreaks in people and horses, sometimes leading to serious brain-related illness and death. This study aimed to investigate WNV circulation in birds and horses in Madrid, Spain. Through serological testing (cELISA), a proportion of birds were positive for WNV antibodies, indicating potential exposure. Four birds and one horse were confirmed positive for WNV antibodies with a second test, while four other birds showed antibodies to an undetermined flavivirus. Because birds positive for WNV antibodies were adults, they might have been exposed elsewhere. The horse had previously lived in a WNV endemic zone. The birds positive for flavivirus included two young birds that had not traveled outside Madrid. The presence of antibodies in two juvenile birds that could hardly fly suggests local circulation of flavivirus in birds in Madrid. The study addresses the potential circulation of WNV or related flaviviruses in birds in Madrid, emphasizing the need for increased surveillance to understand transmission dynamics and the principal species involved. Given the growing incidence and spread of WNV in Spain, continued research is vital for risk assessment and implementing effective control measures.

**Abstract:**

West Nile virus (WNV) is a re-emerging flavivirus, primarily circulating among avian hosts and mosquito vectors, causing periodic outbreaks in humans and horses, often leading to neuroinvasive disease and mortality. Spain has reported several outbreaks, most notably in 2020 with seventy-seven human cases and eight fatalities. WNV has been serologically detected in horses in the Community of Madrid, but to our knowledge, it has never been reported from wild birds in this region. To estimate the seroprevalence of WNV in wild birds and horses in the Community of Madrid, 159 wild birds at a wildlife rescue center and 25 privately owned equines were sampled. Serum from thirteen birds (8.2%) and one equine (4.0%) tested positive with a WNV competitive enzyme-linked immunosorbent assay (cELISA) designed for WNV antibody detection but sensitive to cross-reacting antibodies to other flaviviruses. Virus-neutralization test (VNT) confirmed WNV antibodies in four bird samples (2.5%), and antibodies to undetermined flavivirus in four additional samples. One equine sample (4.0%) tested positive for WNV by VNT, although this horse previously resided in a WN-endemic area. ELISA-positive birds included both migratory and resident species, juveniles and adults. Two seropositive juvenile birds suggest local flavivirus transmission within the Community of Madrid, while WNV seropositive adult birds may have been infected outside Madrid. The potential circulation of flaviviruses, including WNV, in birds in the Madrid Community raises concerns, although further surveillance of mosquitoes, wild birds, and horses in Madrid is necessary to establish the extent of transmission and the principal species involved.

## 1. Introduction

West Nile virus (WNV, *Orthoflavivirus nilense*) is a small positive-stranded, enveloped RNA virus [1] belonging to the genus *Orthoflavivirus* in the viral family *Flaviviridae*. It is a re-emerging zoonotic virus and an arthropod-borne virus, typically transmitted by Culicine mosquitoes, as part of an enzootic cycle in which many bird species can act as reservoirs, or amplifying hosts, and other species, including humans and horses, which are incidental (dead-end) hosts [2]. Typically, the virus undergoes a closed cycle between wild birds and mosquitoes, with birds potentially carrying the virus during their movements, which range from local to long distance [3].

The genus *Orthoflavivirus* comprises more than 70 taxa of viruses, all transmitted by arthropods, with two major clades vectored by mosquitoes or ticks, respectively, a clade with no-known arthropod vector, and a small number of tentative species. In most cases, the primary hosts of the genus *Orthoflavivirus* are mammals or birds. Infections range from asymptomatic to severe or fatal hemorrhagic fever or neurological disease. The most significant human pathogens in this group include yellow fever virus, dengue virus, Japanese encephalitis virus (JEV), tick-borne encephalitis virus (TBEV), WNV, and Zika virus. WNV belongs to the well-supported JEV group, which includes 10 taxa [4,5].

WNV was isolated from a febrile woman from Uganda in 1937 [6] and was known only from Africa, Eurasia, and Australia [4] prior to emerging in New York in 1999 [2]. Currently, WNV is present in numerous countries worldwide, including all countries neighboring Spain, i.e., France [7], Portugal [8], and Morocco [9], where cases have been reported since the 1960’s. In Spain, cases of WNV have been detected in humans, horses, and wild birds. The first human case of WNV in Spain was retrospectively diagnosed from a 2004 sample [10]. Since 2010, human and horse WNV cases have been recovered from Andalusia and Extremadura, Catalonia, and the Valencian Community. The largest outbreak, from Andalusia and Extremadura, in 2020, led to 77 infections and eight deaths [11]. Human cases have been recovered annually since then, with six from Andalusia in 2021, five cases from Andalusia and Catalonia in 2022, and 17 cases from Andalusia, Extremadura, Catalonia, Castilla-La Mancha, and the Valencian Community in 2023 [12]. The first equine outbreak in Spain was reported in 2010, and new cases have occurred annually since then [13]. Moreover, seropositive samples from asymptomatic horses were recovered in central Spain in 2012, including Madrid, where one IgM-positive case was reported, which implied recent infection [14]. 

WNV has been found to infect over 392 distinct bird species [15]. The first, indirect, evidence of WNV transmission in wild birds in Spain, in 2003, was provided by a serosurvey of Common Coot (*Fulica atra*), a partially migratory aquatic bird, in Doñana National Park (Huelva, Southern Spain) [11]. WNV has since been detected in wild [16,17] and captive birds [18,19] in Spain by both direct and indirect surveillance techniques. 

Phylogenetic studies of WNV have shown that there are at least eight distinct lineages [5,20], with different biogeographical patterns. Among them, lineages 1 (L1) and 2 (L2) exhibited the broadest geographical distribution and have been implicated in major outbreaks affecting both humans and animals. Both lineages circulate in Europe [21,22], with L2 increasingly detected in Central and Eastern Europe since 2004, in humans, birds, and horses [21], where most current human cases of WNV were caused by L2 [20]. 

In Spain, WNV L1 is responsible for the majority of WNV infections in humans, horses, birds, and mosquitoes [23]. It is the only lineage detected in the south–west of Spain [24], particularly in Andalusia, with additional cases reported in Extremadura. Rare cases have been notified in Madrid and neighboring regions, such as Ávila (Castile and León), Ciudad Real, and Toledo (Castilla-La Mancha). A secondary, smaller focus of WNV activity has been described in Catalonia, north–eastern Spain, and more recently, in the neighboring Valencian Community [12]. In contrast to the rest of Spain, where L1 has been detected, L2 has been found in Catalonia, specifically in birds since 2017 [25,26], and in mosquitoes in 2021 [27]. WNV disease cases in horses and humans, lineage unknown, have been detected in the north–east since 2022 [28]. The emergence of L2 in the north–east of Spain aligns with a westward dispersion from central Europe [26]. 

Two phylogenetically related *Orthoflavivirus* taxa that show serological cross-reactivity with WNV have been detected in Spain [29]. These include the potentially zoonotic Usutu virus (USUV), a JEV serogroup virus that has an overlapping distribution with WNV in south–west and north–east Spain, and its circulation has also been detected in birds and mosquitoes [30,31,32], and the Bagaza virus (BAGV), a virus in the Ntaya virus serocomplex known from gamebirds in south–west Spain [33]. Circulation in Spain of other more distantly related flaviviruses, such as Meaban virus [34], TBEV, and Spanish sheep/goat encephalitis virus, has been described, but expected cross-reactivity with common WNV serological tests is lower as they belong to distant serocomplexes [32,35]. 

Although WNV has shown to be spreading through the south and north–east of the country, to the best of our knowledge, the only evidence of WNV circulation in the Community of Madrid was the detection of specific antibodies in horses in 2012 [14]. The aim of the present study was to verify the possible circulation of WNV in wild birds and equines in Madrid between 2020 and 2022. Furthermore, several factors that could provide insights into the extent of virus exposure among these animals in this area were examined.

## 2. Materials and Methods

This study consisted of a retrospective study, conducted on wild birds, and a prospective study on equines in order to verify the possible circulation of WNV in these animal species in the Community of Madrid. Serum samples were used in all cases. Institutional ethics clearance was obtained from Alfonso X El Sabio University Ethics Committee to conduct this study (Decision 2022_1/117). 

### 2.1. Samples 

#### 2.1.1. Birds

Blood samples were collected in tubes without anticoagulant from 159 birds in different areas of the Community of Madrid by the GREFA wildlife rescue center (Grupo de Rehabilitación de la Fauna Autóctona y su Hábitat) between August 2020 and August 2021. The field sampling areas covered most of the Madrid Community and were divided into the following regions: Metropolitan area and Henares corridor, Upper Manzanares Basin, Sierra Oeste, Henares Basin, Jarama Basin, Upper Guadarrama Basin, Guadarrama Basin, Sierra Norte, Comarca Sur, and Comarca de Las Vegas (Figure 1). Blood was extracted from the cubital vein or the medial metatarsal vein using 23 G needles. The volume of blood collected varied according to the size of the bird, between 1 and 1.5 mL and never exceeding 1% of the bird’s body weight. Serum was obtained by centrifugation of the samples and then stored at −80 °C for further analysis.

Different variables were collected from each bird, including species, migratory status, gender, age, size, area of origin, cause of admission to the center, subsequent situation/evolution, and presence of pathology or signs of disease at the time of admission. 

Seventeen bird species were sampled, predominantly white storks (*Ciconia ciconia*) with 95 samples, followed by 17 red kites (*Milvus milvus*), 10 Bonelli’s eagles (*Aquila fasciata*), nine griffon vultures (*Gyps fulvus*), five black kites (*Milvus migrans*), four peregrine falcons (*Falco peregrinus*), four Eurasian eagle owls (*Bubo bubo*), three Spanish imperial eagles (*Aquila adalberti*), three black storks (*Ciconia nigra*), two golden eagles (*Aquila chrysaetos*), one great cormorant (*Phalacrocorax carbo*), one grey heron (*Ardea cinerea*), one lesser black-backed gull (*Larus fuscus*), one European honey buzzard (*Pernis apivorus*), one short-toed snake eagle (*Circaetus gallicus*), one great bustard (*Otis tarda*) and one common kestrel (*Falco tinnunculus*). 

Birds were routinely sampled by GREFA personnel as part of standard activities for a variety of purposes, including relocation of nestlings, population monitoring, and treating cases of trauma, malnutrition, natural disease, poisoning, or ingestion of foreign objects.

Birds without sexual dimorphism were either sexed by DNA in an external laboratory if necessary for a specific project or left as “undetermined” if the sex was not relevant in the clinical management or destiny of the animal. The cost and invasiveness of additional tests were taken into consideration when deciding whether to determine the sex. 

Birds were classified as juveniles (including the subcategories nestling, fledgling, and juvenile) or adults based on their feather type. Birds without down, neossoptilus, pre-down, or pre-feather at the time of sample collection were considered adults [36]. Additionally, physical-anatomical characteristics were used to determine age [37].

The size of the bird was determined based on pre-established criteria, considering the bird’s wingspan, and categorized as either large or very large [38].

#### 2.1.2. Equines

Twenty-five serum samples were collected from clinically healthy equids, consisting of twenty-three horses and two donkeys, from different equestrian centers and private properties in the Community of Madrid, between January and April 2022 (Figure 1). Sample inclusion criteria were equids that: (1) did not present clinical symptoms; (2) resided in the Community of Madrid and had not traveled outside the Community at least three months prior to sample collection; and (3) were not vaccinated at any time against WNV. In all cases, informed consent was obtained and signed by the owners of the equids included in the study.

A minimum of 2 mL of blood was obtained in tubes without anticoagulant and subsequently centrifuged to obtain the serum and transported to the Biological Research Unit of the UAX Veterinary Hospital (UIB-UAX). Serum samples were stored at −20 °C until analyzed. 

Different data were collected on the origin, sex, age, and species or breed of the animals.

### 2.2. Serological Testing

Serological testing for the detection of WNV antibodies was conducted using the commercially available competition Enzyme Linked ImmunoSorbent-Assay kit (INgezim WNV Compac enzymatic assay kit, Ingenasa, Spain). Each ELISA plate used for testing included duplicate samples and was read at a wavelength of 450 nm. Results were classified as positive when the inhibition percentage was equal to or greater than 40%. Conversely, results were classified as negative when the inhibition percentage was equal to or less than 30%. Results falling within these two intervals were considered doubtful. Positive cELISA samples from horses were forwarded to the regional reference laboratory (Laboratorio Central de Veterinaria, Algete) for IgM detection (IDVET, France, and Ingenasa, Spain). 

To assess the specificity of antibodies present in the samples, the virus-neutralization test (VNT) was performed on cELISA-positive samples containing sufficient serum (12 birds and 1 horse) and doubtful samples (15 birds and 1 horse) at the Animal Health Research Centre (CISA-INIA), CSIC, following Llorente et al. [29]. Due to the possibility of cross-reactivity with antigenically related flaviviruses documented to be present in Spain, samples were tested in parallel against WNV (strain E101), USUV (SAAR-1776), and BAGV (Spain/RLP-Hcc1/2010) (GenBank accession nos. AF260968, AY453412, and KR108244, respectively). Samples yielding positive neutralization (complete absence of cytopathic effects at dilutions equal to or higher than 1:10) were scored as positive. The presence of specific antibodies for a virus was assigned when titers obtained for that virus were at least 4-fold higher than the others. If the titer differences did not reach this threshold, the sample was scored as “undetermined” flavivirus. 

### 2.3. Statistical Analysis

A descriptive statistical study was conducted, considering all variables collected from each animal. The qualitative variables are presented with their frequency distribution. Positive WNV antibodies in birds identified using cELISA were calculated as a percentage and their respective 95% confidence intervals (95% CI) were determined using the exact binomial method. The statistical analyses were performed using Stata version 13.

## 3. Results

### 3.1. Birds

In the birds examined from Madrid, the percentage of serum samples yielding positive results by WNV cELISA was 8.2% (13/159; 95% CI: 4.4–13.6%). Fifteen avian samples that produced weak inhibition and were considered doubtful were submitted for VNT testing alongside the cELISA-positive samples. Of the twenty-seven bird serum samples analyzed by VNT, four that were positive for cELISA exhibited specific antibodies for WNV (14.8% of the samples challenged by VNT), while four (14.8%), three of them positive and one considered doubtful by cELISA, were categorized as undetermined flavivirus (Table 1). None of the samples were classified as USUV or BAGV positive (presence of specific antibodies). cELISA and VNT titers of the birds that tested positive for VNT are included in Table 2. Municipalities with positive results for WNV are highlighted in Figure 2.

The four WNV-infected birds in Madrid included Bonelli’s eagle (*Aquila fasciata*), black kite (*Milvus migrans*), European honey buzzard (*Pernis apivorus*), and short-toed snake eagle (*Circaetus gallicus*), all belonging to the family Accipitridae (Table 3). These birds were all sampled in different regions of the Madrid Community, namely the Sierra Oeste, Metropolitan area, Sierra Norte, and Guadarrama Basin, respectively. Except for Bonelli’s eagle, all were migratory species. All four WNV seropositive birds were at least 12 months old, though the Milvus migrans had not yet attained definitive adult plumage. The primary reason for admission to the rescue center for the majority of the individuals tested was trauma, except for Bonelli’s eagle, which was captured, marked, and released in the field without requiring passage through the facilities of the rescue center. The four birds exposed to undetermined flaviviruses consisted of three species, i.e., one white stork (*Ciconia ciconia*), family Ciconiidae; two Bonelli’s eagles (*Aquila fasciata*); and one griffon vulture (*Gyps fulvus*), both family Accipitridae. Two of these flavivirus-positive birds were juveniles tested in Madrid on 28 September 2020 and 3 June 2021. 

### 3.2. Equids

One of 25 equid samples collected in Madrid tested positive for WNV antibodies using cELISA screening, showing a prevalence of 4.0% (95% CI: 0.1–20.3%). This horse also tested positive by VNT for WNV. However, IgM detection by the reference laboratory yielded negative results for acute infection. This was a male Anglo-Arab horse whose sample was obtained in the Guadarrama Basin. A second sample, which produced weak inhibition and was considered doubtful for cELISA, was VNT negative.

## 4. Discussion

The main objective of this study was to determine the possible circulation of WNV in birds and horses in the Community of Madrid by detecting antibodies against this virus. Additionally, we aimed to analyze variables that could help us understand the exposure of these animals to the virus in central Spain. Moreover, 8.2% of the birds and 4% of the equines tested yielded positive results for a cELISA test, which is noted for its high sensitivity but is cross-reactive with other related flaviviruses [29]. Seroprevalence findings in birds from this study fall within the range of previous research in Spain using the cELISA tests, indicating seroprevalence rates ranging from 1.96% to 31.25% [39]. Serological results in wild birds from around the world using various cELISA protocols also reveal a broad range of seroprevalence rates: 1.3% in Cyprus [40], 5.5% in Senegal [41], 6.6% in the United States [42], 11.9% in Bangladesh [43], 13.3% in Poland [44], 27.33% in Iran [45], and 32.1% in Romania [46]. 

The WNV seroprevalence of 2.5% recovered by VNT (with a further 2.5% positive for undetermined flavivirus) falls in the lower bound of the range observed for VNT results in birds in Spain of other VNT results in Spain, ranging from 0.66% of the birds originally screened by cELISA to 18.23% [39]. Furthermore, several wild bird studies in Spain show seroprevalence values of 4.1% [30], 4.2% [39], 7.4% [47], and 9.4% [17] for undetermined flaviviruses using VNT. This reflects the challenge of the specific determination of flavivirus antibodies caused by cross-reactions between different assayed viruses. Additionally, we cannot rule out the presence of additional flavivirus taxa.

Similarly, a wide range of seroprevalence values have been reported from equines by VNT, spanning 2.47% in Lebanon [48], 9.8% in Spain [32], 31% in Morocco [49], and 35.7% in Nigeria [50]. 

To the best of our knowledge, WNV exposure has been detected in at least 67 avian species in Spain so far, including all the species that tested positive in this study (Appendix A). The birds that showed specific WNV antibodies by VNT were all adults. Three of them (short-toed snake eagle, European honey buzzard, and black kite) were long-distance migrants and may have been infected in Spain, or equally plausibly, in the sub-Saharan wintering areas to which they migrate. One individual, a Bonelli’s eagle, belongs to a species resident in Spain. However, daily movements of this species are large, especially for juveniles; for instance, one juvenile male tagged with a GPS was recorded to travel 350 km in 48 h [51]. This individual was first captured as an adult in Madrid Province in 2019 and tagged with a GPS locator, and was then recaptured twice in 2020. This individual did not leave Madrid after it was tagged. Unfortunately, no sample was collected at the first time of capture, so it was not possible to determine if the individual was infected in Madrid. In this case, WNV transmission may have occurred in Madrid or in contiguous areas with very similar environmental characteristics to this region. Two hatch-year birds were seropositive for undetermined flavivirus, and WNV infection was neither ruled out nor confirmed for these two individuals. One was a nestling white stork, removed to GREFA as part of a nest relocation scheme prior to the first flight. This seropositive bird must have been infected in its nest in Alcobendas, a conurbation with nearly 120,000 inhabitants [52]. The second bird seropositive for an undetermined flavivirus, a griffon vulture, was captured as a fledgling on the 28th of September. This individual could barely fly and could not have traveled beyond the confines of the Madrid Province. It must have been infected in Madrid. The two adult birds, one resident and one migrant, seropositive for an undetermined flavivirus are not informative in terms of the location of viral transmission, as they have probably traveled outside Madrid province.

The presence of maternal antibodies to West Nile virus has been reported in wild birds, typically persisting for less than 30 days [53,54,55]. One study found that maternal antibodies usually persist for 28 days post-hatching in domestic chickens but exceptionally last for 56 days [56]. The white stork sampled in early June 2021 in Madrid was judged to be considerably older than 30 days due to its large size and weight, and thus is unlikely to possess maternal antibodies. Griffon vultures have a very long period before fledging from their nest of over 120–140 days [57]. This individual was sampled in late September 2020 and is likely to have been at least 150 days old, which is too long for the continued maintenance of maternal antibodies. Similarly, the presence of seroconversion to flavivirus in these birds that have not left Madrid and are too old to maintain maternal antibodies implies circulation within Madrid in 2020 and 2021. 

All the bird species seropositive for WNV or undetermined flavivirus in this study are classified as “least concern” by the International Union for the Conservation of Nature (IUCN). This means none of these species are evaluated as being at risk of extinction at a global level. However, the global population of Bonelli’s eagle is less than 50,000 individuals and is considered to be declining [58]. In Spain, the species is classified as “vulnerable”. All four adult Bonelli’s eagles tested were seropositive by cELISA, and one and two individuals, respectively, were positive for specific WNV or undetermined flavivirus antibodies. This species may be especially at risk from flavivirus infection. However, it is unclear if West Nile or other flavivirus are lethal to this species or impact it in other ways, for instance, in breeding success. In addition, European honey buzzards are classified as “near-threatened” in Spain, with an estimated population of about 1850 individuals and a low population in Madrid [59]. Only one individual of this species was tested, so data are sparse for drawing any conclusions.

All seropositive birds in this study, and indeed all birds tested, were classified as large or very large birds. It has been observed that prevalence of antibodies is directly related to bird size, with higher prevalence found in larger birds [60]. Since we did not test small birds, we cannot directly comment on this observation. It is probable that larger birds are highly attractive to vectors [60]. 

All four birds with specific antibodies to WNV were raptors from the family Accipitridae, supporting the previous observation of the high susceptibility to the infection by the virus in raptors, indicating their important role as reservoirs and spreaders of the virus [61].

Regarding the seropositive result found in a horse in this study, the confirmation from the reference laboratory indicated a negative IgM result. IgM antibodies, indicative of recent infection, typically persist in horses for approximately 1–2 months [62], while IgG antibodies can remain detectable for over a year [13,63]. Considering the sampling period of this horse (January 2022), it is plausible that the horse might have been infected during the preceding months, coinciding with the activity of the transmitting vector (from May to October) [64]. However, this animal resided in a WN-endemic area of the south of Spain two years before its relocation to the Madrid Community; therefore, we could not discard that it was an imported case in the Madrid Community, especially given the absence of outbreaks reported in horses in the region. 

Most WNV cases from humans, horses, and birds in Spain were reported from the south–west, chiefly from Andalusia and Extremadura, with rare cases notified from Madrid neighboring regions. A second, smaller, focus of activity has been described more recently in the north–east of Spain, in Catalonia, and later in the neighboring Community of Valencia [12]. Short WNV sequences were isolated from two golden eagles and a Bonelli’s eagle at a rehabilitation and breeding center in Castilla-La-Mancha, south–central Spain, 2007 [16]. WNV cases have been detected in Toledo, Ciudad Real, and Avila [14,65]. These three locations are 80–180 km from Madrid, and the environmental conditions are similar to those found in Madrid Province. There is thus no obvious barrier to the transmission of WNV in Madrid Province.

This work provides the first evidence for the transmission of flavivirus in Class Aves in Madrid Province in 2020 and 2021. Transmission of this undetermined flavivirus must have occurred by early June, the date of first detection in a hatch-year bird in the Alcobendas municipality. Furthermore, this study underlines that wild birds have been exposed to WNV within traveling distance of Madrid, probably in areas that share similar environmental characteristics. This is concerning since it has been observed that WNV frequently re-occurs in the same area in subsequent years after a first outbreak is reported [26]. The conditions for WNV maintenance and circulation, known to exist in certain areas of the Iberian Peninsula, especially in the south–west, seem to be increasing, consistent with the observation that climate change may favor the establishment of competent mosquito populations in a broader area [13]. Increased surveillance of wild birds, sentinel horses, and insect vectors in Madrid Province and throughout Spain is of utmost importance to better characterize the virus, its behavior, and spread in order to implement appropriate measures to protect public health in Madrid, the third largest conurbation in western Europe. Moreover, characterizing the West Nile virus lineage or lineages potentially present in the Community of Madrid would be essential for risk planning and is especially intriguing as both lineages 1 and 2 are known from Spain.

## Figures and Tables

**Figure 1 vetsci-11-00259-f001:**
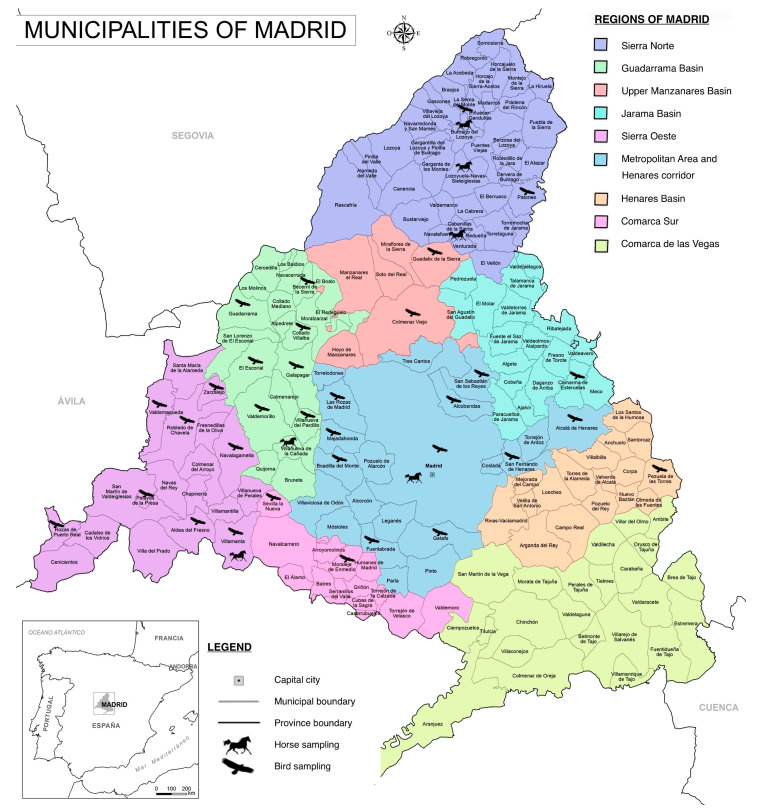
Geographic distribution of bird and horse sampling sites across the Community of Madrid municipalities (reprinted/adapted with permission from netmaps.net, 2016, [©netmaps.net]. Original work derived from BDLJE CC-BY 4.0 ign.es.).

**Figure 2 vetsci-11-00259-f002:**
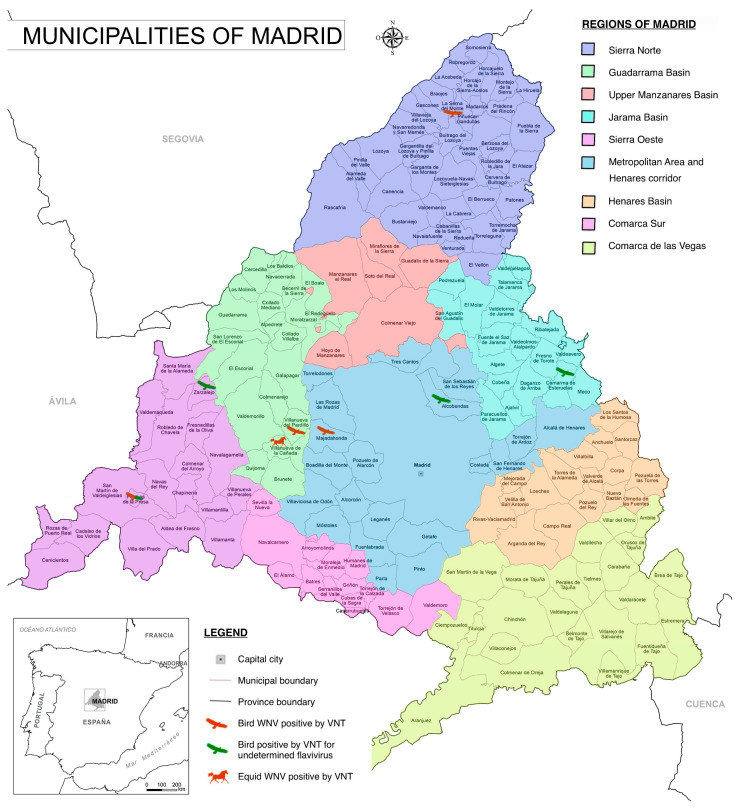
Municipalities in the Community of Madrid, highlighting the locations where birds and equids tested positive by VNT for West Nile virus (WNV) or for undetermined flavivirus antibodies (reprinted/adapted with permission from netmaps.net, 2016, [©netmaps.net]. Original work derived from BDLJE CC-BY 4.0 ign.es.).

**Table 1 vetsci-11-00259-t001:** Species of birds and equines tested for flavivirus antibodies against WNV by capture competitive enzyme-linked immunosorbent assay (cELISA) and virus-neutralization tests (VNT).

					cELISA			VNT			
Order	Family	Species	Common Name	*N*	Pos	Doubt.	Neg	*N*	Undet. Flavivirus	WNV	Neg
BIRDS											
Accipitriformes	Accipitridae	*Aquila adalberti*	Spanish imperial eagle	3			3				
		*Aquila chrysaetos*	Golden eagle	2			2				
		*Aquila fasciata*	Bonelli’s eagle	10	4	1	5	5	2	1	2
		*Gyps fulvus*	Griffon vulture	9	1	5	3	6	1		5
		*Circaetus gallicus*	Short-toed snake eagle	1	1		0	1		1	0
		*Milvus migrans*	Black kite	5	3	1	1	3		1	2
		*Milvus milvus*	Red kite	17		3	14	3			3
		*Pernis apivorus*	European honey buzzard	1	1		0	1		1	0
Charadriiformes	Laridae	*Larus fuscus*	Lesser black-backed gull	1			1				
Ciconiiformes	Ciconiidae	*Ciconia ciconia*	White stork	95	3	3	89	6	1		5
		*Ciconia nigra*	Black stork	3			3				
Falconiformes	Falconidae	*Falco peregrinus*	Peregrine falcon	4			4				
		*Falco tinnunculus*	Common kestrel	1			1				
Otidiformes	Otididae	*Otis tarda*	Great Bustard	1			1				
Pelecaniformes	Ardeidae	*Ardea cinea*	Grey heron	1			1				
Strigiformes	Strigidae	*Bubo bubo*	Eurasian eagle owl	4		1	3	1			1
Suliformes	Phalacrocoracidae	*Phalacrocorax carbo*	Great cormorant	1		1	0	1			1
				158	13	15	131	27	4	4	19
EQUINES											
Perissodactyla	Equidae	*Equus caballus*	Horse	23	1	1	21	2		1	0
		*Equus asinus*	Donkey	2			2				

Abbreviations: cELISA: competitive enzyme-linked immunosorbent assay; VNT: virus-neutralization test; N: total number; Pos: positive; Neg: negative; Doubt: doubtful; Undet: undetermined; WNV: West Nile virus.

**Table 2 vetsci-11-00259-t002:** Titers obtained for all virus-neutralization test (VNT)-positive birds and equines, showing the neutralization against West Nile (WNV), Usutu (USUV), and Bagaza (BAGV) antigens.

	cELISA		VNT			
Common Name	Competition %	Result	WNV	USUV	BAGV	Result
BIRDS						
Bonelli’s eagle	48.54	Positive	1:10	1:20	<1:10	Undetermined flavivirus
	73.25	Positive	1:10	<1:10	<1:10	Undetermined flavivirus
	87.2	Positive	1:20	1:5	1.5	WNV
Griffon vulture	35.17	Doubtful	1:10	<1:10	<1:20	Undetermined flavivirus
European honey buzzard	90.98	Positive	1:40	1:10	<1:10	WNV
Black kite	91.27	Positive	1:160	1:40	<1:10	WNV
White stork	50.54	Positive	1:10	<1:10	<1:10	Undetermined flavivirus
Short-toed snake eagle	No data	Positive	1:80	1:10	<1:10	WNV
EQUINES						
Horse	94.74	Positive	1:320	1:20	<1:10	WNV

Abbreviations: cELISA: competitive enzyme-linked immunosorbent assay; VNT: virus-neutralization test; %: percentage; WNV: West Nile virus; USUV: Usutu virus; BAGV: Bagaza virus.

**Table 3 vetsci-11-00259-t003:** Details on the age, migratory status, sex, location, and zone of all birds found to be VNT positive in this study. Migratory status is shown in parenthesis for one nestling and one fledgling bird, since the former never left its nest and the latter had only flown very short distances.

Common Name	VNT Result	Age	Migratory Status	Sex	Location	Zone
Bonelli’s eagle	Undet. flavivirus	Adult	R	Male	Zarzalejo	Sierra Oeste
	Undet. flavivirus	Adult	R	Male	Pelayos de la Presa	Sierra Oeste
	WNV	Adult	R	Female	Pelayos de la Presa	Sierra Oeste
Griffon vulture	Undet. flavivirus	Fledgling	R	Undet.	Camarma de Esteruelas	Jarama Basin
European honey buzzard	WNV	Adult	M	Undet.	La Serna del Monte	Sierra Norte
Black kite	WNV	Adult	M	Female	Majadahonda	Metropolitan area
White stork	Undet. flavivirus	Nestling	M	Undet.	Alcobendas	Metropolitan area
Short-toed snake eagle	WNV	Adult	M	Male	Villanueva del Pardillo	Guadarrama Basin

Abbreviations: Undet.: undetermined; R: resident; M: migrant.

## Data Availability

The original contributions presented in the study are included in the article/Appendix A; further inquiries can be directed to the corresponding author(s).

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
