# Peer review of "West Nile Virus Seroprevalence in Wild Birds and Equines in Madrid Province, Spain"

_vetsci, 2024, doi:10.3390/vetsci11060259_

Round 1
Reviewer 1 Report
Comments and Suggestions for Authors
This is an interesting study aimed to investigate the activity of West Nile virus through serological techniques within the comuna of Madrid, both in birds and equids. The study is well designed and the authors clearly discuss the meaning of their findings. I recommend acceptance. A few comments below, which I think may improve the English in some sections, and a few questions for the authors aimed to add some additional clarifications in the methods.
Line 62 : …“the” genus..
Line 62: …zoonotic “disease”…
Line 65: …”which are“ incidental (dead-end) hosts…
Line 76:… “WNV belongs to the well supported JEV serogroup”
Line 77…” WNV was isolated from a febrile woman from Uganda in 1937”
Line104-105: In Spain, WNV L1 is responsible for the majority of WNV disease infections in humans, horses, 104 birds, and mosquitoes.
Line 109: in the north-east Spain, in Catalonia..… Replace by “in Catalonia, northeastern” Spain
Line 117: “…such as the Usutu virus (USUV)…
Line 119: “…and the Bagaza virus…”
Line 133-134: Where it says “…Institutional ethics clearance was obtained to conduct this study (Decision 2022_1/117)…” Please, indicate which institution provided this.
Line 138: I think it means “collected”, instead of “corrected”
Lines 145-146: What was the volume of blood collected form these birds?
Figure 1 and 2: I will recommend increasing the font size for the legends indicating the different areas of Madrid and the “comunas”, for easier reading. I suggest to use one, single, larger icon (bird or equine) rather than more than one. They may be too small in the printed or definitive version of the figure.
Lines 164-165: Were all these birds free-ranging admitted to the rehab center or were birds born at GREFA also included? According to this sentence, some appear to be captive bred: “…Birds were routinely sampled by GREFA personnel as part of standard activities for a variety of purposes, including relocation of nestlings, transfers, breeding programs for captive-born animals, ….” If so, please, clarify which ones were included from captive breeding programs…
What was the selection criteria to include some of these birds in this study? Were they opportunistically sampled?
Line 197: Ingezim or INgezim?
Lines 273-275: I think this should be in the discussion.
Line 326-327: This sentence needs clarification: “..The two adult birds, one resident and one migrant, seropositive for an undetermined flavivirus are not informative in terms of transmission…” Unsure about the meaning of “not informative in terms of transmission”.
Line 347: Replace “mortal” by “lethal.”
Comments on the Quality of English Language
Line 62 : …“the” genus..
Line 62: …zoonotic “disease”…
Line 65: …”which are“ incidental (dead-end) hosts…
Line 76:… “WNV belongs to the well supported JEV serogroup”
Line 77…” WNV was isolated from a febrile woman from Uganda in 1937”
Line104-105: In Spain, WNV L1 is responsible for the majority of WNV disease infections in humans, horses, 104 birds, and mosquitoes.
Line 109: in the north-east Spain, in Catalonia..… Replace by “in Catalonia, northeastern” Spain
Line 117: “…such as the Usutu virus (USUV)…
Line 119: “…and the Bagaza virus…”
Line 138: I think it means “collected”, instead of “corrected”
Figure 1 and 2: I will recommend increasing the font size for the legends indicating the different areas of Madrid and the “comunas”, for easier reading. I suggest to use one, single, larger icon (bird or equine) rather than more than one. They may be too small in the printed or definitive version of the figure.
Line 197: Ingezim or INgezim?
Line 347: Replace “mortal” by “lethal.”
Author Response
AUTHORS: Thank you for your valuable feedback and insightful comments on our manuscript. We really appreciate your interest in our work, and we sincerely express our gratitude for your valuable time and diligent effort in evaluating this manuscript. We have made the suggested changes in the manuscript to enhance its quality. We hope that all the concerns have been addressed and that the manuscript has significantly improved after incorporating all the changes suggested. You can find our responses in the attached document.

Reviewer 2 Report
Comments and Suggestions for Authors
The present work describes a serological investigation carried out in Spain and concerning West Nile in wild birds and domestic horses. The study has scientific rigor, the experiments are carried out with correct experimental design, the conclusions obtained are in line with the results obtained. The work is very well written. The sections are well divided. The manuscript is easy to read. I suggest only a few minor comments:
Line 28: Please, delete "a type of"
Simple summary: In the simple summary there are no references to data obtained in equids
Line 50: What about the positive horse? Has the horse been confirmed in VNT?
Line 66: I suggest the authors include a sentence about the detection of antibodies against WNV in other "sentinel" species that have been described over the years. (for example wild boar and hunting dogs in campnia region, pigs in Malasya, Cats Cordoba). I suggest the authors to improve the introduction with these references.
Line 137: How was the number of birds to be tested calculated? And what about horses? Was it convenience sampling?
Comments on the Quality of English Language
English is fine, only minor errors were detected
Author Response
AUTHORS: Thank you for taking the time to review our manuscript and for providing such insightful feedback. We deeply value your effort and attention to detail. In response to your suggestions, we have revised the manuscript to improve its overall quality. We believe that the changes we have made address your comments effectively and have strengthened the manuscript accordingly. Attached you can find the detailed responses to the comments.

Reviewer 3 Report
Comments and Suggestions for Authors
This study is well conducted. However, a few significant points need to be addressed:
Major points:
This study examined the prevalence of WNV in wild birds and horses in Madrid province in Spain. The study focused on seroprevalence using ELISA and VNT. This study is of significance and should be published however, it will benefit from the following changes:
Major comment:
1) The study would benefit from WNV lineage identification specifically since lineages and their significance are being mentioned throughout the manuscript but were never tested.
2) The introduction is lacking: the two clades of Orthoflavivirus should be introduced. Chikungunya virus should also be mentioned as an important zoonotic virus of the genus.
Author Response
AUTHORS: Thank you for dedicating your time and effort to review our manuscript. We really appreciate your input. We have addressed your comments and hope that these revisions meet the reviewers' concerns, making the paper ready for publication. Enclosed you can find the document with the responses to your suggestions.
